# Is Computed-Tomography-Based Body Composition a Reliable Predictor of Chemotherapy-Related Toxicity in Pancreatic Cancer Patients?

**DOI:** 10.3390/cancers15174398

**Published:** 2023-09-02

**Authors:** Marco Cefalì, Isabel Scala, Giuliana Pavone, Daniel Helbling, Saskia Hussung, Ralph Fritsch, Cäcilia Reiner, Soleen Stocker, Dieter Koeberle, Marc Kissling, Vito Chianca, Filippo Del Grande, Sara De Dosso, Stefania Rizzo

**Affiliations:** 1Istituto Oncologico della Svizzera Italiana (IOSI), Ente Ospedaliero Cantonale (EOC), 6500 Bellinzona, Switzerland; marco.cefali@eoc.ch (M.C.); giuliana.pavone@eoc.ch (G.P.); 2Facoltà di Scienze Biomediche, Università della Svizzera Italiana, Via Buffi 13, 6900 Lugano, Switzerland; isabel.scala@usi.ch (I.S.); filippo.delgrande@eoc.ch (F.D.G.); stefania.rizzo@eoc.ch (S.R.); 3Onkozentrum Zürich, Seestrasse 259, 8038 Zurich, Switzerland; daniel.helbling@ozh.ch; 4Department of Medical Oncology and Hematology, University Hospital of Zurich, 8091 Zurich, Switzerland; saskia.hussung@usz.ch (S.H.); ralph.fritsch@usz.ch (R.F.); 5Institute for Diagnostic and Interventional Radiology, University Hospital of Zurich, 8091 Zurich, Switzerland; caecilia.reiner@usz.ch (C.R.); soleen.stocker@usz.ch (S.S.); 6Oncology Departement, St. Claraspital, Kleinriehenstrasse 39, 4058 Basel, Switzerland; dieter.koeberle@claraspital.ch; 7Radiology Department, St. Claraspital, Kleinriehenstrasse 39, 4058 Basel, Switzerland; marc.kissling@claraspital.ch; 8Istituto di Imaging della Svizzera Italiana (IIMSI), Ente Ospedaliero Cantonale (EOC), Via Tesserete 46, 6900 Lugano, Switzerland; vito.chianca@eoc.ch

**Keywords:** body composition, pancreatic cancer, toxicity

## Abstract

**Simple Summary:**

Malnutrition and changes in body composition, such as weight loss and sarcopenia, are frequent in pancreatic cancer patients and are associated with worse survival outcomes according to several studies; however, research has not univocally determined whether or not they are specifically associated with a higher likelihood of toxicity from chemotherapy. This study retrospectively evaluated chemotherapy-related toxicity in a cohort of patients with metastatic pancreatic cancer and explored its relationship with body composition parameters including radiological measurements performed with a specialized software on CT scan images. Statistical analysis failed to show a clear and clinically significant association between the evaluated parameters and chemotoxicity, suggesting that relevant confounding factors likely play a more significant role in determining prognosis.

**Abstract:**

Background: Malnutrition, loss of weight and of skeletal muscle mass are frequent in pancreatic cancer patients, a majority of which will undergo chemotherapy over the course of their disease. Available data suggest a negative prognostic role of these changes in body composition on disease outcomes; however, it is unclear whether tolerance to chemotherapeutic treatment is similarly and/or negatively affected. We aimed to explore this association by retrospectively assessing changes in body composition and chemotherapy-related toxicity in a cohort of advanced pancreatic cancer patients. Methods: Body composition was evaluated through clinical parameters and through radiological assessment of muscle mass, skeletal muscle area, skeletal muscle index and skeletal muscle density; and an assessment of fat distribution by subcutaneous adipose tissue and visceral adipose tissue. We performed descriptive statistics, pre/post chemotherapy comparisons and uni- and multivariate analyses to assess the relation between changes in body composition and toxicity. Results: Toxicity risk increased with an increase of skeletal muscle index (OR: 1.03) and body mass index (OR: 1.07), whereas it decreased with an increase in skeletal muscle density (OR: 0.96). Multivariate analyses confirmed a reduction in the risk of toxicity only with an increase in skeletal muscle density (OR: 0.96). Conclusions: This study suggests that the retrospective analysis of changes in body composition is unlikely to be useful to predict toxicity to gemcitabine—nab-paclitaxel.

## 1. Introduction

Pancreatic cancer (PC) remains the most lethal among malignancies originating from the gastroenteric tract, with 495,773 new cases and 466,003 deaths worldwide in 2020. It is the 12th malignancy by incidence and the 7th by lethality overall [1]. Among PC patients, the majority are diagnosed with locally advanced or metastatic disease, in which a systemic therapy is indicated for palliative purposes, whereas only 20% of them have localized disease eligible for surgery with curative intent [2].

In addition to the traditional PC prognostic indicators (i.e., tumor size, lymph node metastases, surgical margin status, biochemical tumor markers, and adjuvant chemotherapy), malnutrition and subsequent changes in body composition are emerging as factors associated with worse survival outcomes [3,4].

Changes in body composition have been shown to correlate with prognosis in several cancer subtypes, including ovarian, lung, bladder, and pancreatic malignancies [5,6,7,8]. Notably, malnutrition in PC is very common, affecting 30% to 65% of patients [9], and weight loss represents one of the early symptoms that can precede the diagnosis by months in almost 40% of them [10,11].

In these patients, alterations in nutritional status have a multifactorial etiology that includes a paraneoplastic syndrome that affects up to 80% of them, and a malabsorption due to pancreatic exocrine insufficiency (PEI), observed in 44.5–68.0% of cases (whether primary or secondary to previous surgical resection) [12,13]. Indeed, fecal elastase deficiency, as an indicator of PEI, has been identified as an independent predictor of survival [3,14,15]. These factors contribute to the modification of body composition, resulting in a sarcopenia-cachexia syndrome which is characterized by substantial weight loss with a specific loss of skeletal muscle mass [10], which is known to correlate with worse prognosis regardless of the stage of disease [16,17,18].

In some cancer types, sarcopenia is known to increase the toxicity of chemotherapy [19,20], likely because anticancer drug dosing is often based on the global body surface area (BSA) but does not consider the relative distribution between fat and lean mass. Consequently, sarcopenic patients tend to receive a higher dose of chemotherapy compared to a relatively small lean muscle mass and are more prone to suffer drug-related toxicity [21,22].

Several studies have reported poorer responses to chemotherapy and worse overall survival outcomes among sarcopenic PC patients, but a clear correlation with chemotherapy-related toxicity has not yet been demonstrated [23,24,25,26]. However, on the one hand, the impact of changes in body composition on PC patient outcomes has been investigated and available data confirm a negative correlation with overall survival [27]; on the other hand, the literature evidence is not univocal in the definition of the role of sarcopenia in chemotherapy tolerance [28].

Indeed, our group recently performed a systematic review of the available literature on this subject, essentially showing that the association between body composition and chemotherapy-related toxicity in PC is still uncertain [29]. A part of the problem may lie in the lack of a single, univocal definition of a measurable parameter to characterize this syndrome and thus to evaluate its impact—which brings difficulty in establishing a benchmark across studies. Moreover, while the weight or body surface measurement is straightforward, quantitative body composition imaging is not as readily available, and it may require specialized software [15,30,31].

Imaging examinations, including ultrasound, computed tomography (CT) and magnetic resonance, are currently part of the standard of care in pancreatic cancer patients’ assessment for staging and follow-up [32,33,34,35,36].

Therefore, we attempted a multicenter retrospective evaluation of chemotherapy-related toxicities observed in a cohort of metastatic PC patients. The main objective of our study was to demonstrate a correlation between chemotherapy toxicity and body composition measurements, including a CT-based assessment of muscle mass and fat distribution. Specifically, muscle mass was evaluated according to skeletal muscle area (SMA), skeletal muscle index (SMI) and skeletal muscle density (SMD); fat distribution was evaluated by subcutaneous adipose tissue (SAT) and visceral adipose tissue (VAT). The secondary objective was to assess the association between sarcopenia and survival.

## 2. Materials and Methods

### 2.1. Patient Selection

This study’s population was retrospectively selected from a database of patients affected by pancreatic cancer; referred to four different Institutions in Switzerland. The Ethics Committee approved this retrospective study with a waiver for informed consent (2020-01085). Inclusion criteria consisted of age ≥ 18 years; histologically confirmed locally advanced or metastatic pancreatic adenocarcinoma; first-line chemotherapy with gemcitabine and nab-paclitaxel within the last 5 years; and availability in the picture archiving and communication system (PACS) of a CT scan or positron emitting tomography (PET)-CT scan with iodinated-contrast medium performed within 30 days before the start of chemotherapy. Exclusion criteria were concomitant to the diagnosis of other malignancies; loss of a follow-up in the first 6 months after starting the treatment; inadequacy of the CT images due to technical issues, such as the presence of metallic prostheses [37]; and documented refusal to the use of clinical data for research.

### 2.2. Clinical Data Recording

The following clinical data were collected: age at diagnosis; sex; tumor stage; Eastern Cooperative Oncology Group (ECOG) performance status before starting the chemotherapy; body composition values (as defined in the dedicated paragraph); dose reduction of any chemotherapy agent compared to the 1st cycle; cycle delays > 2 weeks due to toxicity; early discontinuation of chemotherapy due to toxicity, defined as treatment termination within two months due to toxicity; occurrence of G3-4 toxicity according to NCI-CTCAE V.4; need for a second-line treatment; blood parameters within 30 days from the date of CT, including hemoglobin; lactate dehydrogenase (LDH); albumin; white blood cells (WBC); and lymphocytes. Weight and height were recorded for the calculation of the body mass index (BMI). Date of the last follow-up, date of disease progression, and death were also recorded. Patients with some missing values were not excluded in order to avoid selection bias.

### 2.3. CT Data Extraction

CT examinations were performed on different CT scanners at different institutions, but they were all available in digital format on our PACS. Feature extraction was performed from the portal venous phase of contrast-enhanced series. An axial image at the level of the third lumbar vertebra (L3) was selected and segmented through the Slice-O-Matic software version 5.0 (Tomovision, Montreal, QC, Canada). The software offers the opportunity to perform either semi-automatic or automatic segmentations, both based on different CT attenuation thresholds for skeletal muscle, SAT and VAT (the automatic segmentation tool is available as adjunctive tool commercially from Voronoi Health Analytics Inc., Coquitlam, BC, Canada; https://voronoihealthanalytics.com (accessed on 3 August 2023)) [38]. Accurate segmentations, checked visually after the software’s use, led to the recording of the following numerical data: skeletal muscle area (including the following muscles: psoas, erector spinae, quadratus lumborum, transversus abdominis, external obliques, internal obliques, and rectus abdominis muscles), measured in centimeters squared; skeletal muscle density (measured by Hounsfield units (HU)); subcutaneous adipose tissue (SAT, expressed in centimeters squared); and visceral adipose tissue (VAT, expressed in centimeters squared). An example of segmentations is shown in Figure 1. The lumbar skeletal muscle index (SMI) was calculated by dividing SMA by square height (m^2^) and reported as cm^2^/m^2^. The sex-specific cut-off to define sarcopenia was SMI < 41 cm^2^/m^2^ for women of any BMI; it was < 43 for underweight and normal weight men; it was < 53 for overweight and obese men [39].

### 2.4. Statistical Analysis

Statistical analyses were performed using STATA16 (StataCorp^®^, College Station, TX, USA). Descriptive statistics were reported as mean and standard deviation, along with the range or relative frequencies and percentages. Pre–post comparisons were performed using the Wilcoxon matched-pairs signed-rank test. Univariate and multivariate logistic regressions were used to assess for toxicity associated with the changes in body composition. Adjustments for age and sex were also performed. The significance level was set at 5% (*p* < 0.05).

## 3. Results

As shown in Table 1, 131 patients (mean age 69.7 ± 9.0 years, 45% women and 55% men), most of whom (81.5%) had metastatic disease at diagnosis, were enrolled in this study. The mean age of this cohort was 69.7 years (42–87). More than 50% of patients had an ECOG score of 1, followed by 34.6% of patients with an ECOG score of 0. At the time of diagnosis, sarcopenia was present in 59 patients (45%), and the mean BMI was 24.2 kg/m^2^. During chemotherapy, 49.2% and 23.4% of patients required a dose reduction and cycle delays, respectively, whereas only 14.8% of patients required an early discontinuation. Grade 3–4 toxicity was noted in approximately 37% of patients based on the criteria defined above. Half of the patients received second-line treatment. The median follow-up time was 10.8 ± 7.8 months (range: 1–44 months).

The median progression-free survival and overall survival were 6 months, 95% CI: 5–7, and 9 months, 95% CI: 8–10, respectively.

The comparison of laboratory tests before and after chemotherapy showed a significant decrease in hemoglobin (pre: 12.5 ± 1.9 g/dL, post: 11.3 ± 2.0 g/dL, *p* < 0.001), albumin (pre: 37.3 ± 5.9 g/L, post: 34.9 ± 5.5 g/L, *p* = 0.001), WBC (pre: 8.0 ± 3.8 K/µL, post: 6.9 ± 4.0 K/µL, *p* = 0.002), and lymphocytes (pre: 1.6 ± 1.1 × 10^3^ cells/µL, post: 1.3 ± 0.7 × 10^3^ cells/µL, *p* = 0.002) (Table 2).

The toxicity risk increased with an increase in SMI (OR: 1.03, 95% CI: 1.02; 1.04) and BMI (OR: 1.07, 95% CI: 1.00; 1.04), whereas it decreased with an increase in SMD (OR: 0.96, 95% CI: 0.95; 0.97). This trend was also confirmed by the univariate logistic regression adjusted for age and sex, with slight variations in OR values (Table 3). Unadjusted and adjusted multivariate analyses confirmed a reduction in toxicity risk only with an increase in SMD (OR: 0.96, 95% CI: 0.95; 0.97).

## 4. Discussion

The combination chemotherapy regimens have been shown to improve overall survival in advanced pancreatic cancer patients [39]. According to the European Society for Medical Oncology (ESMO) guidelines, if the ECOG performance status of the patient is 0 or 1 and the bilirubin level is lower than 1.5× the upper limit of normality (ULN), the possible combination therapies that may be proposed are a triplet combination of 5-fluorouracyl, irinotecan and oxaliplatin (FOLFIRINOX), or a doublet combination of gemcitabine and nab-paclitaxel [40].

At present, there is a lack of data in the literature concerning a direct comparison of FOLFIRINOX and gemcitabine—nab-paclitaxel in pancreatic cancer patients. An indirect comparison performed between the two regimens suggests a slightly greater activity but also a higher toxicity of FOLFIRINOX. Therefore, most patients are treated with nab-paclitaxel in combination with gemcitabine, which represents the preferred first-line regimen due to its better safety profile in comparison to FOLFIRINOX [41,42,43].

While chemotherapy-related adverse events are common, few predictors of toxicity have been identified for clinical use. This plays a strategic role when considering a palliative treatment setting in a lethal disease, such as advanced pancreatic cancer. Chemotherapy-related toxicity negatively impacts quality of life and can also be associated with serious complications, such as febrile neutropenia, that may affect prognosis and require additional medical care or even hospitalization. Being able to identify patients at risk for severe toxicity would allow for appropriate pre-emptive measures, such as modifications in chemotherapy dosage or schedule, and it might also help correct modifiable risk factors. Sarcopenia in pancreatic cancer patients is a well-described entity, and its role as a prognostic factor seems to be quite-well established in pancreatic cancer, but not in other gastrointestinal malignancies [21,28,44,45]. Nonetheless, very little evidence is available regarding its role in predicting chemotherapy toxicity due to the contrasting results of the available data, mostly from retrospective series [30].

In this study, we examined the effects of body composition measurements on treatment-related toxicity. We retrospectively selected a cohort of patients treated homogeneously with nab-paclitaxel in combination with gemcitabine, excluding those who were treated with FOLFIRINOX, in order to minimize confounding factors. These might arise either out of the different toxicity profile of the two combinations, or in terms of treatment allocation, due to the physician’s preference for the triplet in healthier patients.

In our study, the finding of an apparently increased risk of toxicity with the increase in BMI can appear surprising; however, it might be explained by the fact that chemotherapy dosing is commonly calculated on a BSA. Both BSA and BMI are calculated using only height and weight, and do not account for body composition; for instance, a bodybuilder with a high percentage of muscle tissue could have the same BMI as an obese patient [46]. Therefore, the higher toxicity may be due to the BSA-based chemotherapy dose calculation, since high BSA corresponds to a high drug dose which might be disproportionate for an organism with depleted lean mass [47]. Similarly, BSA and BMI do not take into consideration the variations in body compositions that are related to gender and age. Indeed, the aging process is characterized by a decrease in skeletal muscle mass along with a parallel increase in total fat mass, as well as an increase in fat infiltration of muscle and other organs [47]. As for gender, the lower percentage of lean body mass in women compared to men may also represent a confounding factor that neither BSA nor BMI account for [48].

However, it should be cautioned that this interpretation represents an oversimplification of the possible interactions between body composition and drug pharmacokinetics and pharmacodynamics. For instance, the data within the literature are available on the effect of obesity on the pharmacokinetics of different drugs, showing that multiple mechanisms may play a significant role. As discussed by Morrish et al., volume of distribution can change significantly according to protein binding, body composition, and tissue blood flow but the chemical properties of a specific drug will also have to be considered. Moreover, changes in volume of distribution will be more relevant for drugs whose activity is concentration-dependent, and less so for chronically administered time-dependent drugs, where clearance through renal elimination or hepatic metabolism is the predominant factor to determine exposure [49]. Indeed, drug clearance can be influenced by a complex interplay between total body weight, organ weight, and liver and kidney function; for instance, Chagnac et al. showed that, while the glomerular filtration ratio (GFR) is increased in obese subjects compared to average weighed adults, the increase is not linear with body weight [50]. It should be pointed out that data derived from a population of obese patients cannot necessarily be extrapolated to patients of average weight and depleted lean mass.

In our study, the sole positive correlation present at both univariate and multivariate analyses was between SMD and any toxicity, with a higher SMD related to a decreased risk of toxicity. This result is interesting because SMD represents the fatty infiltration of muscle (with higher values corresponding to lower fat infiltration of the muscle fibers), which is considered an indirect estimate of muscle quality. Furthermore, other studies have also demonstrated that SMD is associated with prognosis [51,52,53,54].

The SMI, on the other hand, was significantly correlated with toxicity at a univariate logistic regression, even when adjusted for age and sex. Furthermore, an increase in SMI appeared to correlate with an increase in toxicity, which was unexpected given that a higher SMI corresponds to proportionally higher muscle mass. However, this result was not confirmed at the multivariate analysis, and this discrepancy could suggest that confounding factors are responsible.

In our study, we found that 45% of patients showed sarcopenia at baseline, in line with what has been reported in other cohorts [9]. This condition is, indeed, particularly frequent in pancreatic cancer, possibly due to the activation of the inflammatory response and catabolic pathways. Furthermore, inadequate exocrine function may lead to malnutrition and weight loss [55].

This study has some limitations. The small sample size may be the most impactful, as we recognize that a higher sample size might have been helpful in detecting an association between the CT-based estimate of body composition and chemotherapy-related toxicity. However, the software that was employed for image segmentation and analysis is not routinely utilized in clinical practice, and we deemed it reasonable to explore whether a signal of potential clinical usefulness would emerge in a smaller sample before dedicating resources to a larger study. It was also considered that the involvement of additional institutions, especially beyond national borders, would have added further confounding factors in terms of clinical management.

Moreover, we attempted to balance the homogeneity of this study’s population—and thus the restrictiveness of the inclusion/exclusion criteria—against the sample size. As a consequence, we had to exclude patients who received FOLFIRINOX or gemcitabine monotherapy because the focus of our study was on chemotherapy-related toxicities, and we did not perform subgroup analyses by disease stage (locally advanced versus metastatic) or by other prognostic factors, such as CA 19.9 levels, disease location (pancreatic head versus tail), or laboratory-based scores, since the addition of further variables would generate very small, scarcely-informative subgroups. Further studies are certainly warranted to assess more prognostic factors.

We also recognize that the multicenter and retrospective selection of patients may have increased the risk of selection bias. However, since the selection was decided by each treating oncologist, selection bias was deemed unavoidable. Indeed, because of the retrospective nature of this study, it cannot be excluded that high physician awareness towards this issue of sarcopenia and malnutrition may have contributed to the negativity of our findings. It is not unlikely that visibly sarcopenic patients might have received more nutritional support on the one hand, and more cautious chemotherapy dosing on the other, which could have impacted toxicity. Moreover, the accuracy of the logistic regressions assessed by the ROC curve did not exceed 0.70, indicating a moderate fit of the models in explaining the data. Non-linear relationships between body composition and toxicity were examined without significant results. Furthermore, a 30-day period between a CT scan and blood exams may have been too long; however, given the retrospective selection of patients, we deemed this time window reasonable in the setting of real-world care. In future prospective studies, this time may be made shorter by a pre-inclusion decision. Finally, we did not have nutritional data available about the enrolled patients. These data should also be included in future prospective cohorts.

These unexpected results should be contextualized in the perspective of discrepant data from similar studies, where there is a wide variability of results, with only some experiments demonstrating a significant association between body composition measures and chemotherapy-related toxicity [5,6]. In addition, there is no universal consensus on the ideal cut-off values to define sarcopenia, making a cross-trial comparison not completely trustworthy.

This study suggests that an evaluation of body composition before the start of first-line chemotherapy with nab-paclitaxel and gemcitabine is unlikely to be predictive of toxicity. Nonetheless, a prospective analysis within controlled clinical trials could help to select a more homogeneous population and to obtain reliable and more reproducible data.

## 5. Conclusions

In this retrospective multicentric study, we found a statistically significant association between SMD and any chemotherapy-related toxicity in a retrospective cohort of pancreatic cancer patients treated with nab-paclitaxel and gemcitabine in the first-line setting. However, there was no agreement between SMD and the other body composition parameters that we evaluated, suggesting that multiple confounding factors likely play a more relevant role in determining chemotherapy-related toxicity and overall prognosis. Consequently, it appears unlikely that the evaluation of body composition would be clinically useful for the prediction of chemotherapy-related toxicity at the present time. Larger studies, ideally with a prospective design, may yield more reliable information about this association.

## Figures and Tables

**Figure 1 cancers-15-04398-f001:**
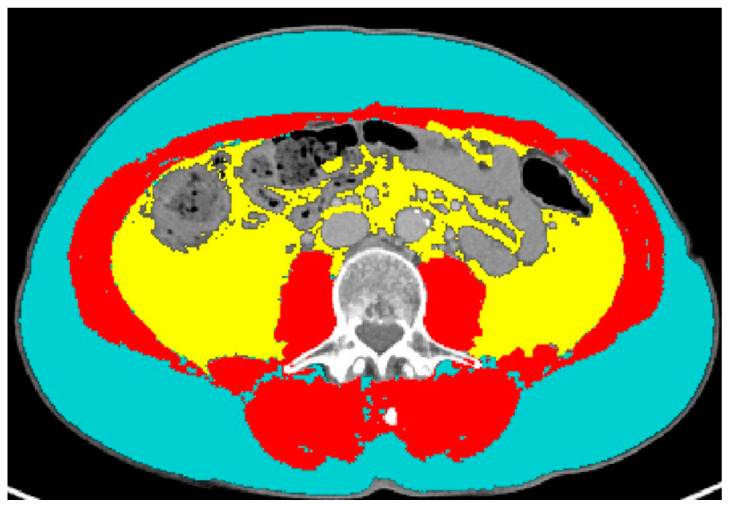
Example of segmentation of subcutaneous adipose tissue (light blue), visceral adipose tissue (yellow) and skeletal muscle area (red).

**Table 1 cancers-15-04398-t001:** Patients’ characteristics.

No. of Patients	131
**Age (years)**	
Mean (SD)	69.7 (9.0)
Range	42–87
**Gender, *n* (%)**	
Female	59 (45.0%)
Male	72 (55.0%)
**Tumor stage, *n* (%)**	
Locally advanced	24 (18.5%)
Metastatic	106 (81.5%)
**ECOG pre, *n* (%)**	
0	45 (34.6%)
1	68 (52.3%)
2	16 (12.3%)
3	1 (0.8%)
**Body composition variables, mean (SD)**	
SAT (cm^2^)	
Mean (SD)	164.1 (81.9)
Range	24.3–390.1
VAT (cm^2^)	
Mean (SD)	137.4 (96.2)
Range	96.2–499.4
SMA (cm^2^)	
Mean (SD)	130.6 (33.8)
Range	36.4–285.2
SMD (HU)	
Mean (SD)	32.6 (14.2)
Range	−8.3–60.7
SMI (cm^2^/m^2^)	
Mean (SD)	45.9 (9.8)
Range	13.5–85.9
Sarcopenia ^1^	59 (45.0%)
BMI, kg/m^2^	
Mean (SD)	24.2 (4.2)
Range	15.2–38.9
**Chemotoxicity, *n* (%)**	
Dose reduction	63 (49.2%)
Cycle delays	30 (23.4%)
Early discontinuation	19 (14.8%)
G3-4 toxicity	43 (37.1%)
Second-line treatment	65 (50.0%)
**Follow-up (months), mean (SD) (range)**	
Mean (SD)	10.8 (7.8)
Range	1–44
**Progression-free survival, *n* (%)**	10 (7.6%)
**Death, *n* (%)**	15 (11.5%)

^1^ Martin cut-off criteria [38]. ECOG = Eastern Cooperative Oncology Group; SD = standard deviation; SAT = subcutaneous adipose tissue; VAT = visceral adipose tissue; SMA = skeletal muscle area; SMD = skeletal muscle density; SMI = skeletal muscle index; BMI = body mass index.

**Table 2 cancers-15-04398-t002:** Laboratory test.

	Pre	Post	*p*-Value
**Hb (g/dL)**			
Mean (SD)	12.5 (1.9)	11.3 (2.0)	<0.001
Range	8.4–20.0	7.6–26.0	
**LDH (U/L)**			
Mean (SD)	373.7 (285.6)	371.5 (161.5)	0.099
Range	101–2636	124–1109	
**Albumin (g/L)**			
Mean (SD)	37.3 (5.9)	34.9 (5.5)	0.001
Range	23–52	17–44	
**White blood count (K/µL)**			
Mean (SD)	8.0 (3.8)	6.9 (4.0)	0.002
Range	2.34–23.4	1.11–35.8	
**Lymphocytes (×10^3^ cells/µL)**			
Mean (SD)	1.6 (1.1)	1.3 (0.7)	0.002
Range	0.15–7.2	0.1–4.7	

**Table 3 cancers-15-04398-t003:** Uni- and multi-variate logistic regressions—outcome: toxicity.

	Univariate OR (95% CI)	Multivariate OR (95% CI)	Univariate Adjusted OR (95% CI)	Multivariate Adjusted OR (95% CI)
**SAT (cm^2^)**	1.00 (1.00; 1.01)	1.00 (0.99; 1.00)	1.00 (1.00; 1.01)	1.00 (1.00; 1.01)
**VAT (cm^2^)**	1.00 (1.00; 1.01)	1.00 (1.00; 1.01)	1.00 (1.00; 1.01)	1.00 (1.00; 1.01)
**SMA (cm^2^)**	1.00 (1.00; 1.01)	1.00 (0.96; 1.02)	1.01 (1.00; 1.02)	1.00 (0.94; 1.04)
**SMD (HU)**	0.96 *** (0.95; 0.97)	0.96 *** (0.94; 0.98)	0.96 *** (0.95; 0.97)	0.96 *** (0.95; 0.98)
**SMI (cm^2^/m^2^)**	1.03 *** (1.02; 1.04)	1.04 (0.93; 1.10)	1.04 *** (1.03; 1.05)	1.05 (0.96; 1.16)
**BMI (kg^2^/m^2^)**	1.07 ** (1.00; 1.14)	0.41 (0.07; 2.32)	1.07 ** (1.01; 1.14)	1.03 (0.93; 1.15)

Significance level: ** = *p* < 0.05, *** *p* < 0.01—logistic regressions adjusted for gender and age—standard errors were determined considering ECOG score as cluster. SAT = subcutaneous adipose tissue; VAT = visceral adipose tissue; SMA = skeletal muscle area; SMD = skeletal muscle density; SMI = skeletal muscle index; BMI = body mass index.

## Data Availability

The data presented in this study are available on request from the corresponding author.

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
