# Peer review of "Is Computed-Tomography-Based Body Composition a Reliable Predictor of Chemotherapy-Related Toxicity in Pancreatic Cancer Patients?"

_cancers, 2023, doi:10.3390/cancers15174398_

Round 1

Reviewer 1 Report

1. Abstarct: there should be not abbreviations in the abstract (SMI - not explained

2. The topic is very interesting ana up-to date - congratulations for authors

3. Introduction: authors stated that prevalence of malnutrition is 30-50%, however in the literature is much higher ca 70% in advanced PC .

4. Methods: in my opinion the interval from CT to biochemistry (30 days) is to long especially in the course of chemotherapy

5. Cut-off for women SMI <41 cm2/m2 what was for men ? I couldn't find

6. It is worth to add commentary that 45 % of patients suffered from sarcopenia 

7. Lack of p value in table 1

8. Lack of information regarding type of nutrition 

Minor spell check is required

Reviewer 2 Report

This paper provides an interesting research topic about therapy-induced toxicity in pancreatic cancer patients.

I would have the following comments and recommendations:

- please use the template of the journal and do not leave the empty pages like now.

- why did you choose only chemotherapy? Radiotherapy can induce increased toxicity in patients. 

- you only mention about a CT scan performed within 30 days before the start of the chemo. When is the second one done? Otherwise, how is the comparison made? 

- how accurate can be your predictions when used in a new cohort of patients? Can you predict the toxicity based on your methods, when there is no indication about it made by the clinician?

-please explain the abbreviations the first time they are used in the text, mostly in tables.

-I think additional references could be helpful for taking into consideration the tumor response to therapy (doi: 10.3390/jcm9061832, doi: 10.1016/j.ctro.2020.06.011, etc).

Please check the English grammar; some phrases do not have punctuation or are too long. The abbreviations have to be mentioned in the text at their first use.

Reviewer 3 Report

1. study volume is too limited; total study volume :131 patients / 5 years, 4 institutions --> about less than 30 patients/ 1 year, 4 institution --> less than 10 patients / 1 year, 1 institution

2. How could you standardized method for estimate body composition from different protocol-based CT scan? Central review? Please be more specific!

3. It would be better to include more specific pancreatic cancer-related clinical data to this analysis: head vs body+tail/ CA 19-9, Prognostic nutritional index (PNI) 

4. mPC and LAPC may be different in terms with body composition. subgroup analysis might be required

5. This study is thought to be important in terms with decision-making process for chemotherapy in pancreatic cancer. However, body composition estimation is not that easy in real clinical practice. Why don't authors corelate body composition with PNI?

English editting is required.

Author Response

Point-by-point response to Reviewer 3 

  1. The study volume is too limited; total study volume :131 patients / 5 years, 4 institutions --> about less than 30 patients/ 1 year, 4 institution --> less than 10 patients / 1 year, 1 institution

Response: We recognize that a higher numerosity might have been helpful in detecting an association between a CT-based estimate of body composition and chemotherapy-related toxicity. However, the software we used for the evaluation of body composition is not routinely employed, and we thought it reasonable to look for a signal large enough to be suggestive of potential clinical usefulness in a smaller sample, before investing resources in a larger study.

We also considered that the involvement of additional institutions, especially if beyond national borders, would have added further confounding factors in terms of clinical management.

We tried not to be restrictive in terms of our inclusion criteria; but at the same time, as the Reviewer rightly observes at points 3 and 4, sample homogeneity needs to be taken into account, and it must be balanced against numerosity. We had to exclude patients who had received FOLFIRINOX or gemcitabine monotherapy, because the focus of our study was on chemotherapy-related toxicity. As for the slow rate of recruitment, it is not relevant in a retrospective study.

  1. How could you standardized method for estimate body composition from different protocol-based CT scan? Central review? Please be more specific!

Response: the software that we used for body composition evaluation estimates the surfaces in cm2 based on the densities on DICOM images. In this study, the segmentation was performed by using the semi-automatic as well the automatic segmentation method, and there was always a visual check of the segmentations done. These segmentations are not strictly dependent on the CT protocol used. Furthermore, in order to make the segmentations more reliable and reproducible, we performed them on the same vascular phase for all the exams (portal venous phase). We added these details and an appropriate reference in the CT data extraction paragraph of the Methods section. Consequently, the other references have been re-numbered.

  1. It would be better to include more specific pancreatic cancer-related clinical data to this analysis: head vs body+tail/ CA 19-9, Prognostic nutritional index (PNI)

Response: we acknowledge that disease heterogeneity (head vs body + tail, CA 19.9 levels, etc.) constitutes a confounding factor. We have had to make choices to balance homogeneity with numerosity; for instance, we limited our study to patients who had received nab-paclitaxel and gemcitabine, excluding other first-line treatment protocols. Considering the limited overall numerosity, the addition of further variables would generate very small subgroups.

  1. mPC and LAPC may be different in terms with body composition. subgroup analysis might be required

Response: As above, the observation is certainly relevant, but considering the small absolute number of patients with locally advanced disease, a subgroup analysis is unlikely to be particularly informative.

  1. This study is thought to be important in terms with decision-making process for chemotherapy in pancreatic cancer. However, body composition estimation is not that easy in real clinical practice. Why don't authors corelate body composition with PNI?

Response: we thank the reviewer for this kind suggestion. We do acknowledge that the PNI is an easy evaluable prognostic score, but it is based on multiplication of albumin and lymphocyte count, therefore is more related to the inflammatory status of the patient [Wang DS, Luo HY, Qiu MZ, Wang ZQ, Zhang DS, Wang FH, Li YH, Xu RH. Comparison of the prognostic values of various inflammation based factors in patients with pancreatic cancer. Med Oncol. 2012 Dec;29(5):3092-100. doi: 10.1007/s12032-012-0226-8] than on quantity and distribution of muscle and fat.

Our study was not dedicated to a large evaluation of prognostic factors in pancreatic cancer. In fact, it was specifically designed to assess associations between specific body composition values, extracted from CT images, and chemotherapy-related toxicities. We did not take into account the ease of this assessment, in order to give priority to the information extracted. Indeed, if the body composition extracted from CT scans would result an important information, the CT vendors might be asked to include specific automatic tools to carry out the segmentation and quantification.

Further studies including PNI are certainly warranted to assess more prognostic factors.

  1. English revision: We have reviewed and improved the text (please see attached)

Round 2

Reviewer 2 Report

All the recommendations have been done in the present manuscript.

Author Response

The Authors would like to thank Reviewers 2 for the final positive evaluation